# The Inhibition Effect and Mechnism of a Thiadiazole Derivative on Q235 Carbon Steel in 1 M HCl Solution

**Yuxin Zhou [1], Ji Tao [1,\*], Dingshuainan Jin [1], Shiping Zhang [1], Yan He [2] and Longlong Niu [1]**

[1]  School of Architecture and Civil Engineering, Nanjing Institute of Technology, Nanjing 211167, China
[2]  School of Civil Engineering, Suzhou University of Science and Technology, Suzhou 215011, China
\*  Correspondence: jitao_9138ji@163.com; Tel.: +86-137-6491-8933

**Abstract:** *N*,*N*-dihydroxyethyl-(5-methyl-[1,3,4] thiadiazol-2-sulfur)-carbonyl acetamide was synthesized and used as an inhibitor to protect Q235 carbon steel in a 1 M HCl solution. The results showed an increased inhibition efficiency with the increase in the concentration of this inhibitor, and an inhibition efficiency higher than 96% at 40 mg/L can be obtained from weight loss, electrochemical impedance spectroscopy, and potentiodynamic polarization results. The inhibition effect was determined by the adsorption film according to the surface morphology and elemental distribution of the carbon steel surface. The adsorption consists of physical adsorption and chemical adsorption in view of the free adsorption energy of −40.64 kJ/mol derived from the Langmuir adsorption isotherm line. The lone pair electrons of N, O and S and π electronics of double bonds in molecules form stable covalent coordination bonds with the empty d orbitals of iron atoms, which is beneficial to chemical adsorption of the inhibitor. The high inhibiton efficiency of this inhibitor is important for the potential application in pickling field.

**Keywords:** thiadiazole derivative; carbon steel; HCl corrosion; electrochemical techniques; adsorption

## 1. Introduction

Mild steel is currently one of the most widely used metal materials in construction, machinery, ships and other fields due to its excellent mechanical properties and low cost [1–3]. However, mild steel is prone to corrosion when it is exposed to corrosive media, such as hydrogen ions, chlorides and carbon dioxide [4–7]. Hydrochloric acid solution is often used to remove impurities and rust on the surface of mild steel in boilers, heat exchangers, oil pipes and other equipment due to its high efficiency and low cost [8]. However, in the acid wash process, hydrochloride solution can also lead to the corrosion of carbon steel, decrease the cross-sectional area, and ultimately shorten the service life of equipment [9].

To solve the above problems, organic inhibitors are usually added to the pickling solution [10–13]. Most organic inhibitors contain hydrophobic non-polar groups and hydrophilic polar groups, which are favorable for forming an adsorption film on mild steel surfaces. The formation of organic films mainly depends on the following physical and/or chemical action [14,15]. The central atom of the organic inhibitor molecule contains lone pair electrons, which can coordinate with hydrogen ions ionized from acidic solution to form cations. These cations will adsorb on the mild steel surface due to electrostatic action, forming a physical adsorption film [14]. This makes the mild steel positively charged, prevents hydrogen ions from contacting the metal surface, increases the activation energy of hydrogen ion discharge, and thereby reduces the corrosion rate of mild steel. The lone pair electrons of N, O, P, S and other atoms in the polar groups of inhibitors can form coordination bonds with the empty d orbitals of iron atoms, so that the inhibitor molecules can form a chemisorption film on the carbon steel surface [15]. In addition, the π electronics of double and triple bonds in some inhibitor molecules can also form coordination bonds with empty d orbitals of iron atoms, and accelerate the formation of chemisorption films on the carbon steel surface [16]. By increasing

the activation energy of the corrosion reaction and decreasing the charge or material transfer related to the corrosion reaction, the corrosion reaction process of the anode and cathode is inhibited by the adsorption film of inhibitor molecules.

Thiadiazole is a five-membered ring compound composed of two carbon atoms, two nitrogen atoms, one sulfur atom and two double bonds. Thiadiazole derivatives can be obtained by substituting the hydrogen atoms linked to the thiadiazole five-membered ring with substituents [17]. Thiadiazole derivatives contain some electro-negative atoms (N, O, P and S) and conjugated systems (benzene and thiadiazole rings) that can provide active electrons for organic adsorption films, making them excellent potential corrosion inhibitors [18–20]. Therefore, they are widely studied and used to protect carbon steel, copper, stainless steel, aluminum alloy and other metals in corrosion media containing hydrochloric acid [21–23], sulfuric acid [24,25], carbonic acid [26–28], sodium chloride [29,30] and so on [31]. For example, Cong et al., synthesized N′-(4-hydroxy-3-methoxybenzylidene)-2-(5-p-tolyl-1,3,4-thiadiazol-2-ylthio) acetohydrazide and used it to protect mild steel in 0.5 M HCl solution. They obtained a high inhibition efficiency of 95% when the inhibitor concentration was higher than 5 mg/L [32]. Lebrini et al., synthesized 2,5-bis(4-pyridyl)-1,3,4-thiadiazole and tested its inhibition effect on mild steel in 1 M HCl solution [33]. They obtained a high inhibition efficiency of 92% when the inhibitor concentration was higher than $2 \times 10^{-4}$ M. Bentiss et al., synthesized six thiadiazole derivatives and utilized them to protect the mild steel in 1 M HCl solution [34]. They found that the inhibition efficiency was related to the electronic properties of thiadiazole derivatives. In summary, the inhibition effect of thiadiazole derivatives on mild steel is mainly determined by the substituents linked to the thiadiazole rings. However, the synthesis route of most thiadiazole derivatives with high anticorrosion efficiency is complex, and most thiadiazole derivatives with simple structures have low anticorrosion efficiency.

Herein, we designed a new thiadiazole derivative, namely, N, N-dihydroxyethyl-(5-methyl-[1,3,4] thiadiazol-2-sulfur)-carbonyl acetamide (TSCA), by substituting the two H atoms with methyl and hydroxyl groups, and synthesized the corrosion inhibitor with a simple two-step synthesis method. The inhibition effect of TSCA on Q235 carbon steel in hydrochloric acid solution was determined by weight loss, electrochemical impedance spectroscopy, and potentiodynamic polarization measurements. The inhibition mechanism of TSCA was analyzed by adsorption isotherms, zero charge potentials and quantum chemical calculations.

## 2. Materials and Methods

### 2.1. Synthesis of TSCA

Initially, 2-thio-5-methyl-1,3,4-thiadiazole, propanedioic acid, and diethanolamine were purchased from Sinopharm Chemical Reagent Co., Ltd., Shanghai and used without purification to synthesize TSCA according to a previous study [35]. The synthesis of TSCA with two steps is illustrated in Figure 1. In the first step, (5′-methyl-[1,3,4] thiadiazole-2′-sulfur-)-carbonyl acetic acid was synthesized. It was used as an intermediate for the synthesis of TSCA. The synthesized TSCA was identified by NMR spectroscopy. 1H NMR (500 MHz, CDCl$_3$), δ: 2.21 (s, 4H, -N-CH$_2$), 2.26 (s, 2H, -CH$_2$), 2.72 (s, 3H, CH$_3$), 2.89 (s, 4H, -O-CH$_2$), 8.89 (s, 2H, -OH).

**Figure 1.** The synthesis route of TSCA.

### 2.2. Preparation of Samples

The Q235 carbon steel used in this work is composed of 0.45% Mn, 0.24% Si, 0.16% C, 0.019% S, 0.0036% P and the remaining Fe in weight. The carbon steel sample used for the weight loss test has a size of $1.50 \times 1.50 \times 3.00$ cm, and that used for the electrochemical test and surface observation has a size of $1.00 \times 1.00 \times 1.00$ cm. All the carbon steel samples were polished with emery papers (grit 400, 1200, and 2000), washed with ethyl alcohol and dried naturally before any test.

The corrosion solution (1 M HCl) in this work was prepared by diluting concentrated hydrochloric acid (37 wt.%) with distilled water. The inhibitor solutions were prepared by dissolving TSCA inhibitor (1, 10, 20, and 40 mg) in 1 L corrosion solution.

### 2.3. Weight Loss Measurement

The weight loss measurement was conducted by immersing the carbon steel samples in the 1 M HCl solution with or without TSCA inhibitor at 298 K. After 3 h, the samples were removed, washed with distilled water, blow dried, and then weighed. The weight loss is calculated as the weight difference of each carbon steel sample before and after immersion.

### 2.4. Electrochemical Experiments

Electrochemical experiments, including open circuit potential (OCP), electrochemical impedance spectroscopy (EIS) and potentiodynamic polarization curves, were performed one after another in a three-electrode electrochemical cell with a CHI 660E electrochemical workstation. In the electrochemical cell system, carbon steel with an exposed surface ($1.00 \times 1.00$ cm) was used as the working electrode, saturated calomel electrode (SCE) was used as the reference electrode, and a platinum plate ($1.00 \times 1.00$ cm) was used as the counter electrode. The three electrodes were immersed in the HCl solutions with different TSCA contents at 298 K.

The OCP measurement was conducted first, and it was stopped when the potential reached a stable state. In the OCP test, the potential value was recorded as $E_{OCP}$ once the potential reached stability (variation less than 2 mV in 300 s). The EIS test was performed with sine wave circuit excitation (amplitude: 10 mV; frequency: $10^5$–$10^{-2}$ Hz) at $E_{OCP}$. The potentiodynamic polarization curve test was performed by changing the potential from $-250$ mV to $+250$ mV versus $E_{OCP}$ at a scan rate of 0.5 mV s$^{-1}$ from the cathodic side to the anodic side.

### 2.5. Quantum Chemical Calculations

The geometrical optimization of the TSCA molecule was performed with Gaussian 09 under B3LYP/6-311++G (d, p) level based on density functional theory (DFT). The relevant quantum chemical parameters were obtained from this optimized structure.

### 2.6. Surface Observation and Elemental Analysis

The carbon steel specimens were exposed to the 1 M HCl solution for 1 h, removed, and then dried naturally before surface observation. The morphology analysis of the carbon steel specimen was carried out with a scanning electron microscope (SEM) (S3400, HITACHI, Tokyo, Japan). The element composition of the exposed carbon surface was analyzed with an energy dispersive spectrometer (EDS) (OXFORD Link-ISIS-300, Oxford, UK).

## 3. Results and Discussion

### 3.1. Weight Loss Measurement

The weight of the carbon steel specimen was weighted before and after immersion in 1 M HCl solution for 3 h. Thus, the corrosion rate of the carbon steel can be calculated from the weight loss, exposed surface area and immersion time. The derived inhibition efficiency from weight loss measurement ($\eta_w$) is shown in the following equation:

$$\eta_w = \frac{W_0 - W_1}{W_0} \times 100\% \tag{1}$$

where $W_0$ and $W_1$ are the corrosion rates (g·m$^{-2}$·h$^{-1}$) of the carbon steel sample after immersion in the 1 M HCl solution without and with the TSCA inhibitor, respectively. The corrosion rate of carbon steel and inhibition efficiency of TSCA at various concentrations for carbon steel are listed in Table 1. The corrosion rate of carbon steel decreases rapidly as TSCA is added into the corrosion solution, and it decreases with increasing TSCA concentration of the corrosion solution. As a result, the inhibition efficiency of TSCA for carbon steel increases with increasing concentration within the scope of this study. This may be caused by the larger coverage of TSCA on the carbon steel surface at higher concentrations [23].

**Table 1.** Corrosion rate of carbon steel and inhibition efficiency of TSCA at various concentrations for carbon steel in 1 M HCl at 298 K.

| $C_{\text{TSCA}}$ (mg·L$^{-1}$) | $W$ (g·m$^{-2}$·h$^{-1}$) | $\eta_{\text{W}}$ (%) |
| --- | --- | --- |
| 0 | 19.08 | - |
| 1 | 6.68 | 65.0 |
| 10 | 2.96 | 84.5 |
| 20 | 1.62 | 91.5 |
| 40 | 0.76 | 96.0 |

### 3.2. Electrochemical Impedance Spectroscopy (EIS)

Figure 2 shows the Nyquist plots of the carbon steel electrode in 1 M HCl solutions with different contents of TSCA at 298 K. The addition of TSCA does not change the shape of the Nyquist plot, indicating that the electrochemical reaction mechanism of carbon steel remains unchanged. However, the radius of the plot increases with increasing TSCA concentration from 0 to 40 mg/L. This reveals that the corrosion process of carbon steel is significantly inhibited by TSCA. All the Nyquist plots are similarly depressed semicircle-shaped in the frequency range from $10^5$ to 0 Hz, demonstrating that the corrosion reaction process of carbon steel is mainly determined by the charge transfer rate [36,37]. The capacitive arcs are flattened due to the coarse exposed surface of carbon steel adsorbing TSCA molecules, ions, and corrosion products [38,39]. In addition, inductive loops arise in the Nyquist plots in the frequency range from 0 to $10^{-2}$ Hz. This is caused by the adsorption and desorption behavior of TSCA [40]. Thus, the obtained EIS experimental data was fitted by the equivalent circuit model, as illustrated in Figure 3. In the model, $R_\text{s}$, $R_\text{ct}$, and $R_\text{L}$ represent the resistance of the corrosion solution, charge transfer, and inductive, respectively. $L$ and $CPE$ represent the inductive and constant phase elements, respectively. The derived inhibition efficiency from EIS measurement ($\eta_\text{E}$) is shown in the following equation:

$$\eta_\text{E} = \frac{R_\text{ct}^1 - R_\text{ct}^0}{R_\text{ct}^1} \times 100\% \tag{2}$$

where $R_{ct}^0$ and $R_{ct}^1$ are the fitted charge transfer resistances in the presence and absence of TSCA inhibitor, respectively.

The fitted impedance parameters and the corresponding inhibition efficiency for carbon steel immersed in 1 M HCl solutions with different contents of TSCA are given in Table 2. The charge transfer resistance of carbon steel and the corresponding inhibition efficiency increase with increasing TSCA concentration, and the highest inhibition efficiency of 96.3% can be obtained at 40 mg/L. This is in accordance with the weight loss result. The double-layer capacitance ($C_\text{dl}$) and deviation parameter ($n$) are derived from the following equation [41]:

$$C_\text{CPE} = \frac{C_\text{dl}}{(2\pi f_\text{max})^{n-1}} \tag{3}$$

where $C_{CPE}$ represents the capacitance of the *CPE*, and $f_{max}$ represents the corresponding frequency of the maximum -$Z''$ value in the Nyquist plot. As shown in the table, the double-layer capacitance decreases with increasing TSCA concentration. This is caused by the change in thickness of the film ($d$) and local dielectric constant ($\varepsilon$), which can be calculated by the following equation [42]:

$$C_{dl} = \frac{\varepsilon^0 \varepsilon A}{d} \tag{4}$$

where $\varepsilon^0$ is the permittivity of air, and $A$ is the exposed surface area. On the carbon steel/solution interface, the TSCA molecules replace the water molecules, thickening the film and decreasing the local dielectric constant in the replacement process due to the larger particle size and lower electrical conductivity of the TSCA molecule than the water molecule [43]. Thus, the double-layer capacitance decreases in the presence of TSCA molecules, which is good for protecting the carbon steel. In addition, the deviation parameter value is controlled by the inhomogeneity of the carbon steel surface. The lower deviation parameter value in the presence of TSCA in the corrosion solution indicates a lower corrosion degree of the carbon steel electrode. In conclusion, TSCA at high concentrations causes a large replacement of water on the carbon steel/solution interface, thereby resulting in a good inhibition effect on carbon steel.

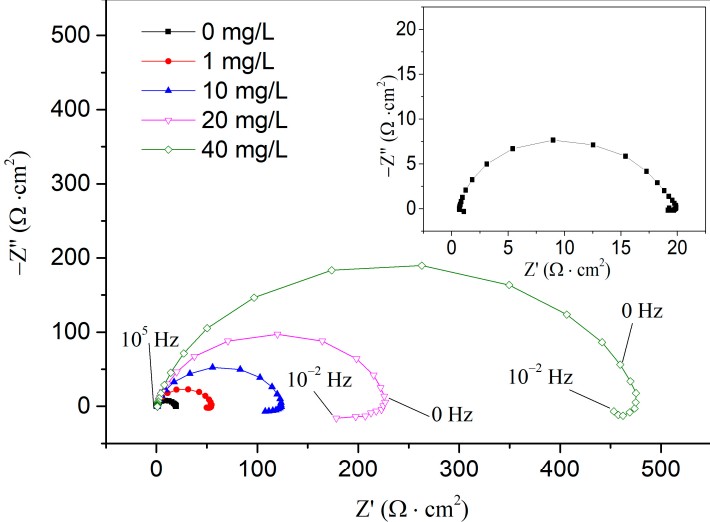

**Figure 2.** Nyquist plots of the carbon steel electrode immersed in 1 M HCl solutions with different contents of TSCA at 298 K.

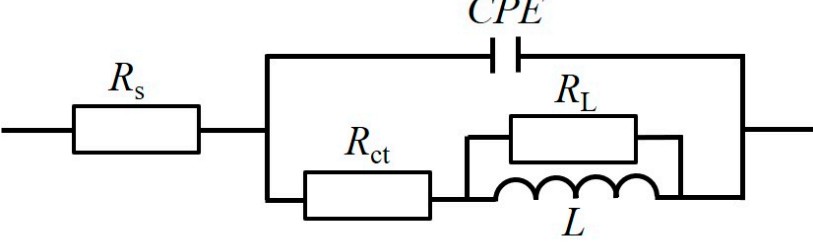

**Figure 3.** Equivalent circuit model used to fit EIS experiment data of carbon steel electrode.

**Table 2.** Impedance parameters and corresponding inhibition efficiency for carbon steel immersed in 1 M HCl solutions with different contents of TSCA.

| $C_{TSCA}$ (mg·L$^{-1}$) | $R_S$ ($\Omega \cdot$cm$^2$) | $C_{dl}$ ($10^6$/s$^n \cdot \Omega^{-1} \cdot$cm$^{-2}$) | $n$ | $L$ (H·cm$^2$) | $R_L$ ($\Omega \cdot$cm$^2$) | $R_{ct}$ ($\Omega \cdot$cm$^2$) | $\eta_E$ (%) |
|---|---|---|---|---|---|---|---|
| 0 | 0.79 | 208.52 | 0.80 | 35.17 | 3.46 | 18.67 | - |
| 1 | 0.79 | 134.26 | 0.93 | 113.57 | 53.22 | 52.05 | 64.1 |
| 10 | 0.84 | 86.88 | 0.92 | 302.42 | 187.53 | 113.61 | 83.6 |
| 20 | 0.94 | 68.23 | 0.91 | 756.41 | 528.90 | 247.73 | 92.5 |
| 40 | 0.77 | 49.83 | 0.91 | 1183.72 | 734.68 | 504.43 | 96.3 |

### 3.3. Potentiodynamic Polarization Curves

Figure 4 represents the potentiodynamic polarization curves of the carbon steel electrode immersed in 1 M HCl solutions with different contents of TSCA at 298 K. It can be observed that the polarization curves move in the direction of decreasing current density in the presence of TSCA. This result demonstrates that the anode and cathode reaction of the carbon steel electrode were inhibited by TSCA in the corrosion solution.

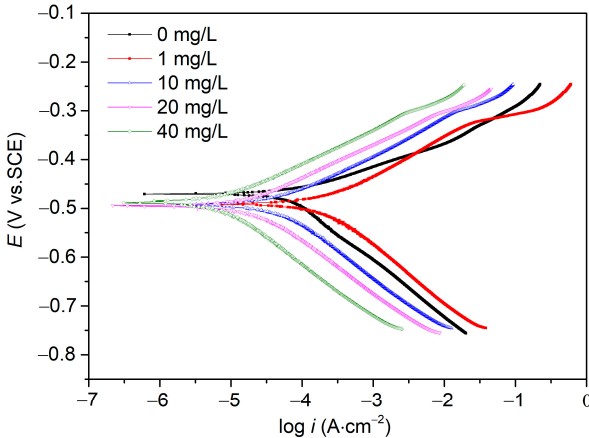

**Figure 4.** Potentiodynamic polarization curves of the carbon steel electrode immersed in 1 M HCl solutions with different contents of TSCA at 298 K.

The polarization parameters, including corrosion potential ($E_{corr}$), corrosion current density ($i_{corr}$), and cathodic and anodic Tafel slopes ($\beta_c$ and $\beta_a$) are derived from the Tafel polarization region as the steel was not corroded via localized corrosion with the formation of pits [44–46]. The results are listed in Table 3. The shift in $E_{corr}$ is less than 10 mV, indicating that TSCA works as a mixed-type inhibitor inhibiting both the anode and cathode electrochemical reactions. In addition, slight shifts in $\beta_c$ and $\beta_a$ can also be observed in the table. This result demonstrates that the electrochemical reaction of carbon steel remains unchanged, which is in line with the EIS results. The decrease in corrosion current density of carbon steel with TSCA concentration increase reveals the inhibition effect of TSCA on carbon steel. The corresponding inhibition efficiencies ($\eta_E$) at different concentrations of TSCA can be calculated by the following equation:

$$\eta_T = \frac{i_{corr}^0 - i_{corr}^1}{i_{corr}^0} \times 100\% \tag{5}$$

where $i_{corr}^0$ and $i_{corr}^1$ are the fitted corrosion current densities of carbon steel in the absence and presence of TSCA inhibitor, respectively. The calculated inhibition efficiencies at different concentrations of TSCA are also listed in Table 3. They are in good accordance with the calculated inhibition efficiencies in weight loss and electrochemical impedance spectroscopy measurements. The highest inhibition efficiency of 97.3% can be obtained at 40 mg/L, demonstrating that TSCA is an effective inhibitor for carbon steel. The increase in inhibition efficiency with increasing TSCA concentration is related to the adsorption

amount of TSCA on the carbon steel surface. A higher adsorption amount of TSCA increases the steric hindrance and thereby insulates the ions from the carbon steel surface [47].

**Table 3.** Polarization parameters and corresponding inhibition efficiency for carbon steel immersed in 1 M HCl solutions with different contents of TSCA.

| $C_{TSCA}$ (mg·L$^{-1}$) | $E_{corr}$ (mV) | $i_{corr}$ (µA·cm$^{-2}$) | $-\beta_c$ (mV·dec$^{-1}$) | $\beta_a$ (mV·dec$^{-1}$) | $\eta_T$ (%) |
|---|---|---|---|---|---|
| 0 | −478 | 846 | 132 | 72 | - |
| 1 | −480 | 284 | 123 | 75 | 66.4 |
| 10 | −483 | 121 | 126 | 73 | 85.7 |
| 20 | −481 | 65 | 129 | 71 | 92.3 |
| 40 | −483 | 23 | 128 | 68 | 97.3 |

*3.4. Adsorption Isotherm Analysis*

　　The inhibition effect of TSCA on carbon steel is determined by the adsorption process from electrochemical impedance spectroscopy and potentiodynamic polarization tests. To study the adsorption of TSCA, the adsorption isotherm in a thermodynamic way is applied considering a good inhibition effect. The surface coverage ($\theta$) is the inhibition efficiency from the weight loss, EIS and potentiodynamic polarization experiment. Langmuir, Temkin, and Frumkin adsorption isotherms are used to study the relationship between the surface coverage and the concretration of TSCA ($C$), as shown in Figures 5–7. The adsorption of TSCA follows the Langmuir adsorption isotherm, as the linear regression coefficient of the Langmuir adsorption isotherm is closer to 1 than that of the Temkin or Frumkin adsorption isotherm from the weight loss and EIS results. In addition, the linear regression coefficients of the Langmuir adsorption isotherm and Frumkin adsorption isotherm are almost the same. So, the adsorption of TSCA follows the Langmuir adsorption isotherm from the results. The Langmuir adsorption isotherm can be expressed by the following equation:

$$\frac{C}{\theta} = C + \frac{1}{K_{ads}} \tag{6}$$

where $K_{ads}$ represents the equilibrium constant for the adsorption/desorption process. The value of $K_{ads}$ ($2.40 \times 10^5$ L/mol) can be calculated from the intercept of the fitted line. It is related to the free adsorption energy ($\Delta G^0_{ads}$), the absolute temperature ($T$), and the gas constant ($R$), as shown in the following equation:

$$\Delta G^0_{ads} = -RT \ln(55.5 K_{ads}) \tag{7}$$

　　A large negative value of –40.64 kJ/mol for $\Delta G^0_{ads}$ was obtained from the equation, indicating a strong interaction between TSCA and carbon steel on the solid/solution interface [48,49]. Since the obtained $\Delta G^0_{ads}$ is less than –40 kJ/mol, the adsorption of TSCA can be ascribed to chemical action. However, the small difference of –0.64 kJ/mol between the obtained true value and reference value reveals physical action between TSCA and carbon steel [50]. As a result, chemical adsorption contributes much to the adsorption of TSCA on the carbon steel surface, while physical adsorption contributes relatively little. The lone pair electrons of N, O and S and $\pi$ electronics of double bonds in TSCA molecules form stable covalent coordination bonds with the empty d orbitals of iron atoms, which is beneficial to the surface coverage and inhibition efficiency of carbon steel.

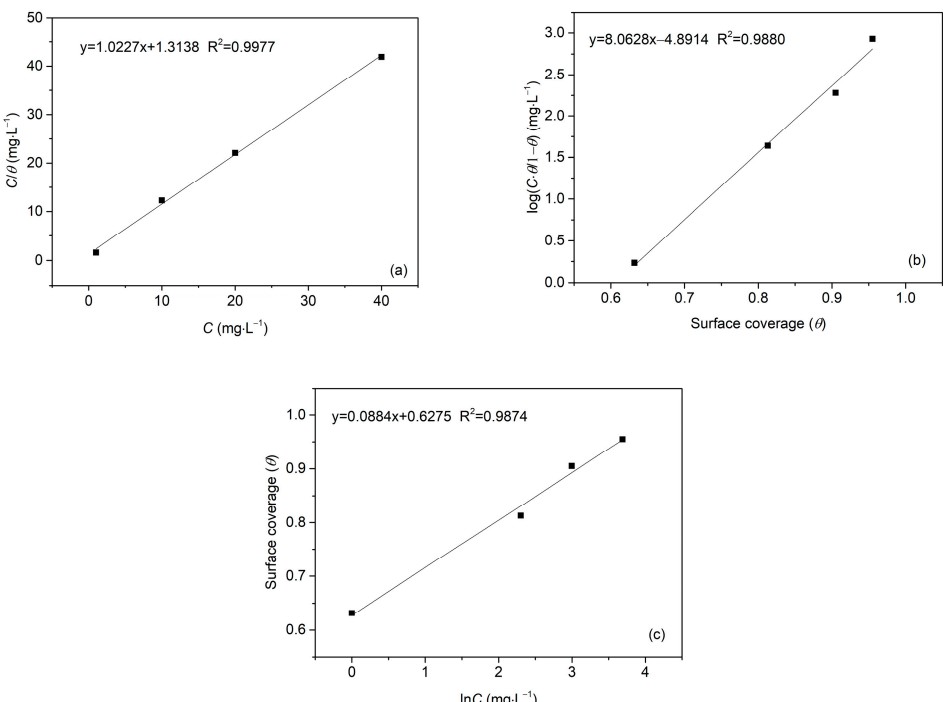

**Figure 5.** Different adsorption isotherm plots for the adsorption of TSCA on the surface of carbon steel in 1 M HCl at 298 K from weight loss results: (**a**) Langmuir's isotherm, (**b**) Frumkin's isotherm, (**c**) Temkin's isotherm.

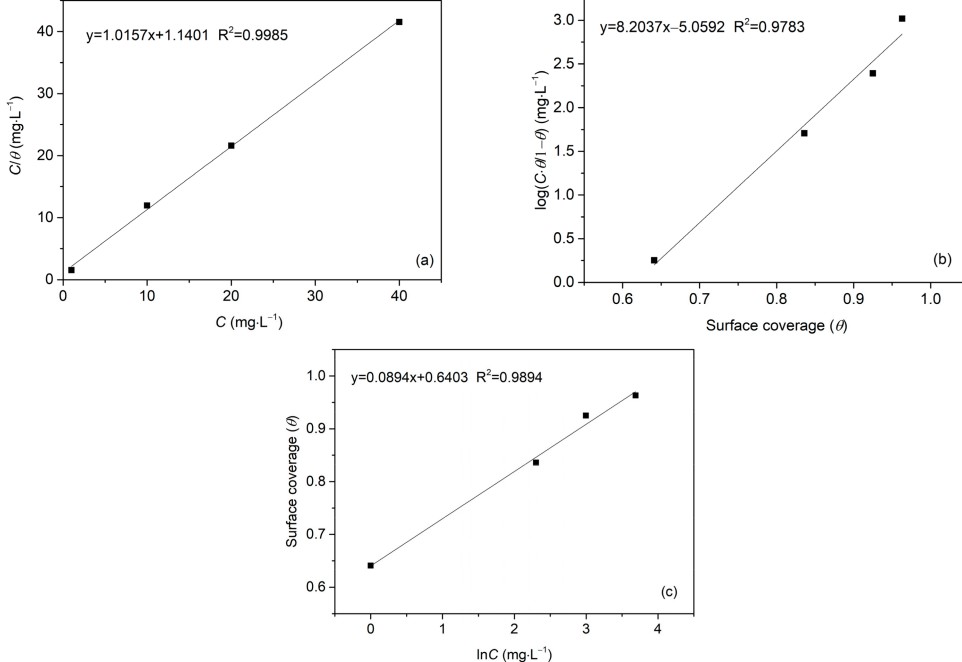

**Figure 6.** Different adsorption isotherm plots for the adsorption of TSCA on the surface of carbon steel in 1 M HCl at 298 K from EIS results: (**a**) Langmuir's isotherm, (**b**) Frumkin's isotherm, (**c**) Temkin's isotherm.

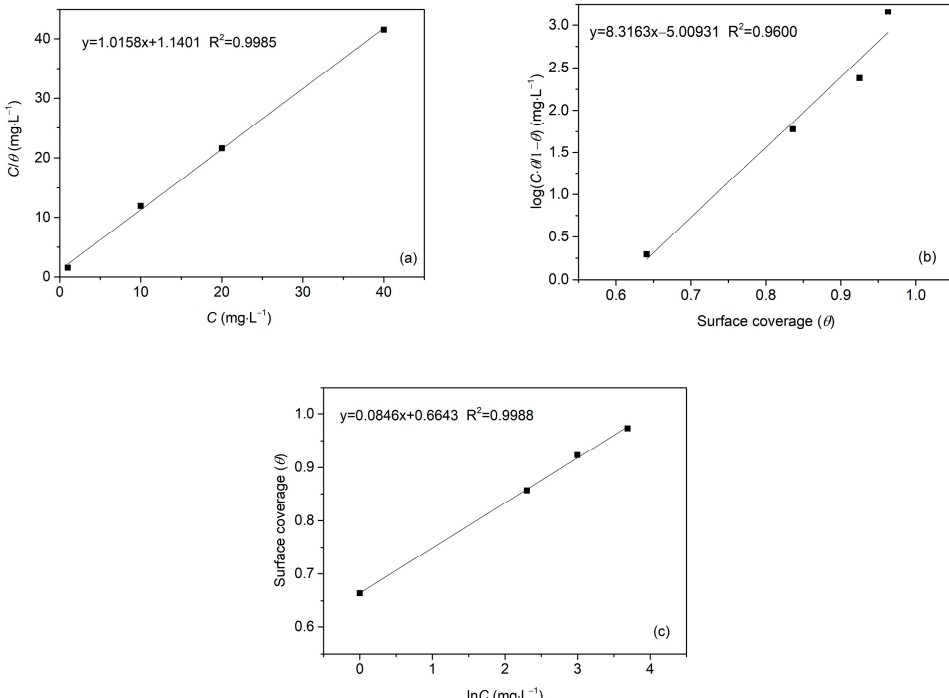

**Figure 7.** Different adsorption isotherm plots for the adsorption of TSCA on the surface of carbon steel in 1 M HCl at 298 K from potentiodynamic polarization results: (**a**) Langmuir's isotherm, (**b**) Frumkin's isotherm, (**c**) Temkin's isotherm.

### 3.5. Quantum Chemical Analysis

The above measurements showed a good inhibition effect of TSCA on carbon steel and indicated that the inhibition effect depends on the adsorption film. To reveal the interaction between TSCA molecules and carbon steel, quantum chemical calculations were conducted with density functional theory. Figure 8 presents the density distributions of the highest and lowest occupied molecular orbitals (HOMO and LUMO) for the TSCA molecule. The highest occupied molecular orbitals are mainly distributed at O=C–N and C–O–H, and the lowest occupied molecular orbitals are mainly distributed at the thiadiazole ring. Table 4 gives the calculated HOMO energy ($E_{HOMO}$), LUMO energy ($E_{LUMO}$), and energy gap ($\Delta E = E_{LUMO} - E_{HOMO}$). Considering the high $E_{HOMO}$ (−6.92 eV), the O=C–N and C–O–H groups can easily donate electrons to iron atoms, which favors chemical adsorption. In addition, the low energy gap of 4.64 eV indicated a high inhibition efficiency of TSCA inhibitor [32].

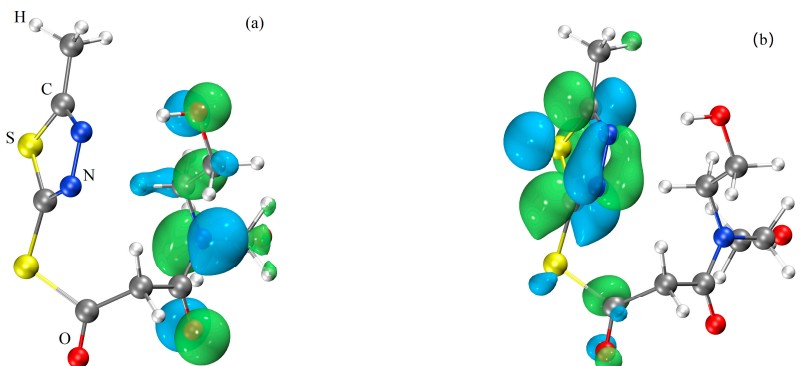

**Figure 8.** The frontier molecular orbital density distribution of the TSCA molecule: HOMO (**a**) and LUMO (**b**).

**Table 4.** Derived quantum chemical parameters for TSCA calculated by the DFT/B3LYP method.

| Name | $E_{HOMO}$ (eV) | $E_{LUMO}$ (eV) | $\Delta E$ (eV) | $\chi$ (eV) | $\gamma$ (eV) | $\Delta N$ |
|------|------|------|------|------|------|------|
| TSCA | −6.92 | −2.28 | 4.64 | 4.60 | 2.32 | 0.65 |

Some other parameters, such as electronegativity ($\chi$), global hardness ($\gamma$), and fraction of electrons transferred ($\Delta N$) can be derived from the following equations:

$$\chi = -\frac{E_{HOMO} + E_{LUMO}}{2} \tag{8}$$

$$\gamma = \frac{E_{LUMO} - E_{HOMO}}{2} \tag{9}$$

$$\Delta N = \frac{\chi_{Fe} - \chi_{TSCA}}{2(\gamma_{Fe} + \gamma_{TSCA})} \tag{10}$$

The theoretical values of $\chi_{Fe}$ and $\gamma_{Fe}$ are 7 eV and 0 eV, respectively [51]. The calculated $\Delta N$ value of 0.65 demonstrates the good chemisorption property and inhibition efficiency of TSCA for carbon steel [52].

### 3.6. Surface Observation and Elemental Analysis

The surface morphology of carbon steel immersed in 1 M HCl solutions without and with 40 mg/L TSCA for 1 h at 298 K is shown in Figure 9. In the absence of TSCA inhibitor, the carbon steel surface is rough due to the cover of corrosion products. In contrast, the surface is comparatively smooth in the presence of TSCA. To demonstrate the adsorption of TSCA, an energy diffraction spectrum experiment was carried out on the exposed carbon steel surface. From the two energy diffraction spectra, it can be found that characteristic peaks of N and S arise in the presence of TSCA. This indicates the adsorption of TSCA on the carbon steel surface. The TSCA film inhibits the corrosion of carbon steel, which is in accordance with the electrochemical experiments and adsorption isotherm results.

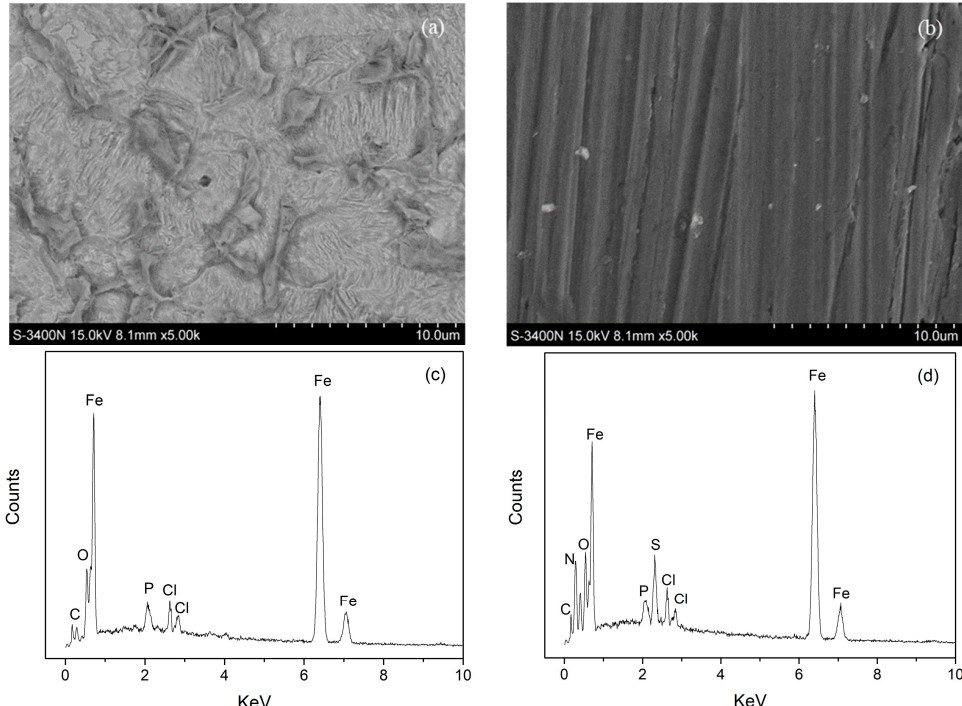

**Figure 9.** SEM micrographs and EDS results of carbon steel immersed in 1 M HCl solutions without TSCA (**a**,**c**) and with 40 mg/L TSCA (**b**,**d**) for 1 h at 298 K.

In conclusion, TSCA effectively inhibited the corrosion of carbon steel in 1 M HCl solution in light of the weight loss, electrochemical impedance spectroscopy, potentiodynamic polarization, adsorption isotherm, potential of zero charge, and surface morphology results. The TSCA film adsorbed on the carbon steel surface inhibited the cathodic and anodic reactions.

## 4. Conclusions

We synthesized a new thiadiazole derivative with a simple two-step synthesis method and utilized it to protect carbon steel in 1 M HCl solution. Comprehensive evaluation of the inhibition effect was conducted with weight loss, electrochemical impedance spectroscopy, and scanning electron microscopy experiments. The calculated inhibition efficiencies from these methods were in good agreement, and they indicated that TSCA can effectively protect carbon steel. The inhibition effect of TSCA was determined by the adsorption film, which can simultaneously reduce the cathodic and anodic reaction rates of carbon steel. According to the adsorption isotherm results, the formation of the TSCA film is due to a combined function of chemical adsorption and physical adsorption with predominantly the first one. In light of the good inhibition effect, easy synthesis, and low cost of TSCA, this compound may be applied in different fields, such as acid pickling, cleaning, and descaling.

**Author Contributions:** Data curation, Y.Z., D.J. and L.N.; funding acquisition, S.Z., Y.H., Y.Z. and J.T.; writing—original draft, Y.Z. and D.J.; writing—review and editing, J.T., S.Z., L.N. and Y.H. All authors have read and agreed to the published version of the manuscript.

**Funding:** This work was financially supported by Science Foundation of Nanjing Institute of Technology (No. YKJ201929), the Natural Science Foundation of the Jiangsu Higher Education Institutions of China (No. 20KJB560005), the Key Project of Jiangsu University Student Practice Innovation Training Program (No. 202111276021Z) and the National Science Foundation of China (No. 51808369 and 52078247).

**Institutional Review Board Statement:** Not applicable.

**Informed Consent Statement:** Not applicable.

**Data Availability Statement:** The data presented in this study are available on request from the corresponding author.

**Conflicts of Interest:** The authors declare no conflict of interest.

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
