# Peer review of "The Inhibition Effect and Mechnism of a Thiadiazole Derivative on Q235 Carbon Steel in 1 M HCl Solution"

_applsci, doi:10.3390/app13042103_

Round 1

Reviewer 1 Report

This manuscript describes an experimental study of the inhibition of Thiadiazole Derivative on Q235 carbon steel in acid solution. The background and context of the work are clearly described and the results are described in suitable detail. In some places I found the manuscript needs to be supplemented and modified. I recommend publication in Applied Sciences following minor revision.

1.       Page 2 line93. It would be perfect if there’s an NMR spectrum result. Can the authors provide this in the manuscript?

2.       P3 line 120. Can the authors give the counter electrode?

3.       Page4 line159, the Frequencies mentioned here should be marked in the Figure2.

4.       Page4 line161-162. The authors wrote that “The capacitive arcs are flattened due to the coarse exposed surface of carbon steel adsorbing TSCA molecules, ions, and corrosion products”. This sentence is not accurate enough.

Refer to the Reference 39  ” The constant phase element (CPE) is used to explain the non-ideal capacitance properties of the metal surface due to the surface heterogeneity. The surface heterogeneity is resulted from surface roughness, the adsorption of inhibitors, and formation of surface layers on the metals”, author can consider rewrite this part.

5.       Can the authors provide the detailed information of RL and L?

6.       The current density in Fig.4 should be A/cm2.

7.       Page 10 line327 .SEM image is a2D image, so it cannot be concluded that if there’re pits. A 3D image is a good way to judge the pits. Author can consider this.

Author Response

I appreciate the chance to further improve this manuscript. And, I sincerely thank the reviewers for the comments from them too.

The following is the comments from the reviewers and the answers from me. The answers are written in blue.

The changes are marked in red in order to facilitate both you and the reviewers to see the modifications I made in this manuscript.

Reviewer 1

This manuscript describes an experimental study of the inhibition of Thiadiazole Derivative on Q235 carbon steel in acid solution. The background and context of the work are clearly described and the results are described in suitable detail. In some places I found the manuscript needs to be supplemented and modified. I recommend publication in Applied Sciences following minor revision.

  1. Page 2 line93. It would be perfect if there’s an NMR spectrum result. Can the authors provide this in the manuscript?

Answer: The synthesis of the corrosion inhibitor was according to Reference 35, and the NMR results were also from Reference 35.

  1. P3 line 120. Can the authors give the counter electrode?

Answer: A platinum plate (1.00 × 1.00 cm) was used as the counter electrode. The description was added in the manuscript.

  1. Page4 line159, the Frequencies mentioned here should be marked in the Figure2.

Answer: The frequencies mentioned was marked in Fig. 2 in the revised manuscript.

  1. Page4 line161-162. The authors wrote that “The capacitive arcs are flattened due to the coarse exposed surface of carbon steel adsorbing TSCA molecules, ions, and corrosion products”. This sentence is not accurate enough.

Refer to the Reference 39  ” The constant phase element (CPE) is used to explain the non-ideal capacitance properties of the metal surface due to the surface heterogeneity. The surface heterogeneity is resulted from surface roughness, the adsorption of inhibitors, and formation of surface layers on the metals”, author can consider rewrite this part.

Answer: The description “The capacitive arcs are flattened due to the coarse exposed surface of carbon steel adsorbing TSCA molecules, ions, and corrosion products” was revised as “The capacitive arcs are flattened because of the heterogeneous carbon steel surface, which adsorbes TSCA molecules, ions, and corrosion products”.

  1. Can the authors provide the detailed information of RLand L?

Answer: The information of RL and L were added in Table 2.

  1. The current density in Fig.4 should be A/cm2.

Answer: The current density in Fig.4 was corrected.

  1. Page 10 line327 .SEM image is a 2D image, so it cannot be concluded that if there’re pits. A 3D image is a good way to judge the pits. Author can consider this.

Answer: We are sorry for the unavailable 3D image of carbon steel. The description of pits was deleted in the revised manuscript.

Reviewer 2 Report

The subject of this study is the corrosion inhibition of carbon steel in 1 M HCl solution by a newly synthesized thiadiazole derivative. The authors use the classical methods of determining the effectiveness and mechanism of inhibition. The article is in accordance with the specialization of the journal, however, the general impression is that certain parts of the manuscript are presented superficially and have certain ambiguities that need to be improved.

- In lines 105 and 106, the preparation of the base corrosion solution (1 M HCl) is given. It is necessary to give details of the inhibitor solutions used.

- Move equation 1 to section 3.1

- Please provide the conditions for performing OCP measurements and show the results obtained in Section 3 (these data are the basis for determining the potential of zero charge).

- It is not clear how the EIS and polarization measurements were made: immediately after immersion in solution or after some stabilization on OCP? Please clarify.

- After what time period were the EIS measurements performed? I ask to list the data for L and RL in Table 2 and highlight the quality of the fitting procedure with the chi-squared element (chi2).

- What is the effect of stabilization time on EIS response (e.g., at the highest inhibitory concentration)?

- Analysis of the adsorption isotherm should be performed for all methods used to determine the inhibitory effect of TSCA (for EIS, potentiodynamic polarization, and weight loss experiments). Such an extended analysis would provide more reliable results on the adsorption of TSCA on carbon steel.

- It is not entirely clear how the potential of zero charge was determined (whether stabilization was performed at a specific potential). I request that in Section 3.5 the results of the EIS measurements be presented graphically (at selected potentials) and in tabular form.

- The description of the quantum chemical analysis is missing in the experimental part. Was a molecular dynamic simulation performed?

- Line 275: "Epzc is defined as the potential at which Rct reaches a maximum...". Why? Please explain.

- I ask to describe in more detail the HSBA principle in corrosion inhibition and explain the influence of parameters c, g and DN on the inhibition mechanism of TSCA on carbon steel.

- What is the effect of TSCA concentration on Epzc?

Author Response

I appreciate the chance to further improve this manuscript. And, I sincerely thank the reviewers for the comments from them too.

The following is the comments from the reviewers and the answers from me. The answers are written in blue.

The changes are marked in red in order to facilitate both you and the reviewers to see the modifications I made in this manuscript.

Reviewer 2

The subject of this study is the corrosion inhibition of carbon steel in 1 M HCl solution by a newly synthesized thiadiazole derivative. The authors use the classical methods of determining the effectiveness and mechanism of inhibition. The article is in accordance with the specialization of the journal, however, the general impression is that certain parts of the manuscript are presented superficially and have certain ambiguities that need to be improved.

- In lines 105 and 106, the preparation of the base corrosion solution (1 M HCl) is given. It is necessary to give details of the inhibitor solutions used.

Answer: The inhibitor solutions were prepared by dissolving TSCA inhibitor (1, 10, 20, and 40 mg) in 1 L corrosion solution.

- Move equation 1 to section 3.1

Answer: Equation 1 was moved to section 3.1.

- Please provide the conditions for performing OCP measurements and show the results obtained in Section 3 (these data are the basis for determining the potential of zero charge).

Answer: The conditions “The three electrodes were immersed in the HCL solutions with different TSCA contents at 298 K. The OCP measurement was conducted first, and it was stopped when the potential reaches a stable state.” were added in the revised manuscript.

- It is not clear how the EIS and polarization measurements were made: immediately after immersion in solution or after some stabilization on OCP? Please clarify.

Answer: The EIS and polarization measurements were made after some stabilization on OCP (variation less than 2 mV in 300 s).

- After what time period were the EIS measurements performed? I ask to list the data for L and RL in Table 2 and highlight the quality of the fitting procedure with the chi-squared element (chi2).

Answer: The EIS measurements were performed once the potential was stable (variation less than 2 mV in 300 s). The data for L and RL were added in Table 2 in the revised manuscript. The EIS fitting was conducted by ZSimpWin software, the quality of the fitting was evaluated by rel. std. errors (less than 10%). We really do not undertand the chi-squared element used to highlight the quality of the fitting procedure.

- What is the effect of stabilization time on EIS response (e.g., at the highest inhibitory concentration)?

Answer: The EIS response was more stable (with less noises) and its fitness by ZSimpWin was more accurate after some stabilization on OCP (variation less than 2 mV in 300 s). The stabilization for EIS test was a traditional operation.

- Analysis of the adsorption isotherm should be performed for all methods used to determine the inhibitory effect of TSCA (for EIS, potentiodynamic polarization, and weight loss experiments). Such an extended analysis would provide more reliable results on the adsorption of TSCA on carbon steel.

Answer: Analysis of the adsorption isotherm based on the EIS and potentiodynamic polarization results was added in the revised manuscript (Fig. 6 and Fig. 7).

- It is not entirely clear how the potential of zero charge was determined (whether stabilization was performed at a specific potential). I request that in Section 3.5 the results of the EIS measurements be presented graphically (at selected potentials) and in tabular form.

Answer: The potential of zero charge was determined by the EIS measurements at different potentials (EOCP ± 10, 20, 30, 40, 50, 60, 70, 80, 90, and 100 mV). The EIS measurements were performed at EOCP after some stabilization on OCP (variation less than 2 mV in 300 s). Then EIS measurements were performed at other potentials immediately after EIS measurements at EOCP. Considering the mistakes in defining and analysing Epzc, we deleted Section 3.5 “Potential of Zero Charge Analysis” in the revised manuscript.

- The description of the quantum chemical analysis is missing in the experimental part. Was a molecular dynamic simulation performed?

Answer: The geometrical optimization of TSCA molecule was performed with Gaussian 09 under B3LYP/6-311++G (d, p) level based on density functional theory (DFT). And relevant quantum chemical parameters were obtained from this optimized structure. The description was added in Section 2.5 in the revised manuscript.

- Line 275: "Epzc is defined as the potential at which Rct reaches a maximum...". Why? Please explain.

Answer: The “Epzc is defined as the potential at which Rct reaches a maximum” is according to Reference 32(Electrochemistry 2015, 83(4), 262–267). But other references (J. Colloid Interface Sci. 2020, 572, 91–106. J. Taiwan Inst. Chem. E. 2019, 96, 588–598.) reveal that “Epzc is defined as the potential at which Rp reaches a maximum”. To avoid the mistake, we deleted Section 3.5 “Potential of Zero Charge Analysis” in the revised manuscript.

- I ask to describe in more detail the HSBA principle in corrosion inhibition and explain the influence of parameters c, g and DN on the inhibition mechanism of TSCA on carbon steel.

Answer: Consdering the mistake in testing Epzc, the Potential of Zero Charge Analysis was deleted in the revised manuscript. Thus, the adsorption of TSCAH+ was not used to describe the inhibition mechanism of TSCA on carbon steel based on the HSAB principle.

- What is the effect of TSCA concentration on Epzc?

Answer: The effect of TSCA concentration on Epzc was not clear, as we just tested the Epzc without TSCA and with 40 mg/L TSCA. We will test the Epzc at different TSCA concentrations and explore its effect in the next work.

Round 2

Reviewer 2 Report

The revised version of the manuscript has been significantly improved. The manuscript can be published in Applied Sciences after correction of minor errors:

- Lines 23 and 125: "... HCL..." should be "....HCl...". I ask the authors to check this also in the rest of the text.

Author Response

I appreciate the chance to further improve this manuscript. And, I sincerely thank the reviewers for the comments.  The mistake "HCL" was corrected in the whole text.